# Pre-training Sequence, Structure, and Surface Features for Comprehensive Protein Representation Learning

**Youhan Lee**[*]**, Hasun Yu**[*]**, Jaemyung Lee**[*]**, Jaehoon Kim**
Kakao Brain
{youhan.lee,shawn.yu,james.brain,jack.brain}@kakaobrain.com

## Abstract

Proteins can be represented in various ways, including their sequences, 3D structures, and surfaces. While recent studies have successfully employed sequence- or structure-based representations to address multiple tasks in protein science, there has been significant oversight in incorporating protein surface information, a critical factor for protein function. In this paper, we present a pre-training strategy that incorporates information from protein sequences, 3D structures, and surfaces to improve protein representation learning. Specifically, we utilize Implicit Neural Representations (INRs) for learning surface characteristics, and name it ProteinINR. We confirm that ProteinINR successfully reconstructs protein surfaces, and integrate this surface learning into the existing pre-training strategy of sequences and structures. Our results demonstrate that our approach can enhance performance in various downstream tasks, thereby underscoring the importance of including surface attributes in protein representation learning. These findings underline the importance of understanding protein surfaces for generating effective protein representations.

## 1 Introduction

Proteins are vital components of biological systems, executing a myriad of functions that underpin an extensive array of cellular processes and biological pathways. These intricate macromolecules have multi-faceted characteristics that can be represented through different paradigms, including but not limited to their amino acid sequences, three-dimensional (3D) structures, and the specific attributes of their surface regions. In recent years, advancements in high-throughput sequencing (HTS) technologies, cryogenic electron microscopy (cryo-EM), and sophisticated algorithms for protein structure prediction (Jumper et al., 2021; Baek et al., 2021; Lin et al., 2022) have led to an explosion of available protein sequence (Suzek et al., 2007) and structure (Berman et al., 2000; Varadi et al., 2022; Lin et al., 2023) data, most of which have been made publicly accessible. Leveraging these abundant datasets, recent studies (Meier et al., 2021; Zhang et al., 2022; 2023) have successfully employed machine learning models pre-trained on this data, resulting in significant progress in tackling an array of downstream tasks in the field of protein science.

Despite these strides, there exists a notable oversight in the current landscape of protein representation learning: the often-underestimated significance of protein surface characteristics. The attributes of a protein's surface are crucial in determining its functional properties, particularly in the context of molecular interactions like ligand binding, enzymatic catalysis, and signal transduction between molecules (Gainza et al., 2020; Somnath et al., 2021). While existing works for protein representation learning have focused heavily on encoding amino acid sequences and 3D structural elements, they have largely neglected the indispensable role that protein surfaces serve, thus leaving an unaddressed gap in the prevailing research. More specifically, the protein structure can be divided into the atoms comprising the backbone and the components constituting the side chains. In this context, the protein surfaces are determined by both backbone and side chain atoms. However, traditional protein structure encoders typically process protein 2D graphs or 3D geometric graphs that only contain

---
[*]These authors contributed equally to this work

Table 1: Comparison of different protein encoders with and without sequence, structure, or surface pre-training. Our model, ESM-GearNet-INR-MC, covers three modalities, sequence, structure, and surface in both encoding and pre-training, achieving comprehensive protein representation learning.

| Method | Sequence Encoder | Structure Encoder | Sequence Pre-training | Structure Pre-training | Surface Encoder | Surface Pre-training |
|---|---|---|---|---|---|---|
| CNN | ✓ | | | | | |
| Transformer | ✓ | | | | | |
| GVP | | ✓ | | | | |
| GearNet | | ✓ | | | | |
| ESM-1b | ✓ | | ✓ | | | |
| ProtBert | ✓ | | ✓ | | | |
| DeepFRI | ✓ | ✓ | ✓ | | | |
| LM-GVP | ✓ | ✓ | ✓ | | | |
| ESM-GearNet | ✓ | ✓ | ✓ | | | |
| GearNet-MC | | ✓ | | ✓ | | |
| GearNet-DP | | ✓ | | ✓ | | |
| ESM-GearNet-MC | ✓ | ✓ | ✓ | ✓ | | |
| ESM-GearNet-INR-MC (Ours) | ✓ | ✓ | ✓ | ✓ | ✓ | ✓ |

alpha carbon or backbone atoms, respectively. Consequently, state-of-the-art representations often lack consideration for side-chain information.

In response to this significant gap, our research aims to offer a comprehensive solution. We propose an all-encompassing pre-training strategy that incorporates information from all three essential aspects of proteins: sequences, 3D structures, and notably, surfaces. Our approach is pioneering in that it is the first to specifically target the learning of protein surface attributes, and it employs cutting-edge Implicit Neural Representations (INRs) (Chen & Wang, 2022) to achieve this goal effectively. This inclusive approach enables our model to enhance performance across various downstream tasks, thereby emphasizing the importance of incorporating surface information in protein representation learning.

In summary, our contributions include:

- We are the first to propose a pre-training strategy that incorporates information from protein sequences, structures, and surfaces.
- We utilize Implicit Neural Representations (INRs) as an effective mechanism for learning surface characteristics of proteins.
- We conduct a comprehensive comparison of the effects of pre-training on protein sequences, structures, and surfaces, thereby demonstrating the efficacy of learning about surfaces.

## 2 RELATED WORK

### 2.1 PROTEIN REPRESENTATION LEARNING

Most studies in the field of protein representation learning have adopted one of three main approaches: (i) focusing on protein sequences, (ii) concentrating on protein structures, or (iii) employing a hybrid strategy that incorporates both sequence and structural information.

In the first approach, which focuses on learning protein sequences, researchers commonly adopt the architecture of pre-trained language models from the field of Natural Language Processing (NLP), such as Transformer(Vaswani et al., 2017), BERT (Devlin et al., 2018), and GPT (Radford et al., 2018), to effectively represent proteins by learning their amino acid sequences as if they were language (Elnaggar et al., 2007; Meier et al., 2021; Notin et al., 2022). The second approach generally employs Graph Neural Networks (GNNs)-based architectures (Ingraham et al., 2019; Jing et al., 2020; Hermosilla et al., 2020; Zhang et al., 2022) to capture the intricate structural features of proteins. In the third approach, hybrid models aim to learn from both protein sequences and structures. Notable studies, such as DeepFRI (Gligorijević et al., 2021) and LM-GVP (Wang et al., 2022) have utilized encoders for both sequence and structural information and have pre-trained on sequence data. STEPS (Chen et al., 2023) and ESM-GearNet (Zhang et al., 2023) have gone a step further by also pre-training on structural information to achieve enhanced performance.

However, these methods have not taken into account the significant role of protein molecular surface information plays in various biological processes. Traditionally, molecular surfaces are defined using Connelly surfaces (Connolly, 1983; Sanner et al., 1996) based on van der Waals (vdW) radii, often represented as mesh-based structures derived from signed distance functions. Seminal work for modeling protein molecular surfaces is MaSIF (molecular surface interaction fingerprinting) (Gainza et al., 2020), which fingerprints molecular surfaces expressed as molecular meshes using pre-defined and pre-calculated physical and geometrical features. To remove the high pre-computation costs of featurization, Sverrisson et al. (2021) proposed dMaSIF, which showed that modeling molecular surfaces as a point cloud with atom categories per point is competitive. Somnath et al. (2021) proposed HOLOProt, which attempted to segment the protein surface into "superpixels" for more efficient consideration of surface information and used the features in conjunction with structure features in a multi-modal modeling manner. However, theoretically, molecular surfaces are continuous surfaces with infinite resolution, which existing mesh-based approaches cannot fully express. To tackle this challenge, we utilize the Implicit Neural Representations (INRs) approach, a technique capable of perfectly capturing infinite resolution characteristics. Our model, called ProteinINR, understands protein molecular surface resolution independently. Furthermore, our ProteinINR model is a generalizable INR approach, allowing us to develop a single model capable of representing many protein structures. On the other hand, Wang et al. (2023) proposed a harmonic message passing, called HMR, which considered surfaces during molecular representation learning. Compared to HOLO-Prot and HMR, which focused on the design of encoder, we use INRs as a pre-training framework in which a structure encoder is trained to extract structure features to recover molecular surface.

## 2.2 IMPLICIT NEURAL REPRESENTATIONS

Point cloud-based (Qi et al., 2017a;b; Thomas et al., 2019; Zhang et al., 2021), mesh-based (Sinha et al., 2016; Bagautdinov et al., 2018; Verma et al., 2018), and voxel-based (Curless & Levoy, 1996; Wu et al., 2015; Tatarchenko et al., 2017; Zeng et al., 2017) methods have historically relied on fixed-sized coordinates or grids to represent 3D assets. Unfortunately, these approaches suffer from resolution dependency, making them insufficient for modeling or rendering high-resolution 3D assets effectively. In contrast, Implicit Neural Representations (INRs) concentrate on learning parameterized functions that predict location-specific information for given arbitrary query coordinates by utilizing seminal methods such as auto-decoding (Park et al., 2019b; Mescheder et al., 2019), Fourier features (Tancik et al., 2020; Mildenhall et al., 2021), sinusoidal activations (Sitzmann et al., 2020b), meta-learning (Tancik et al., 2021; Dupont et al., 2022a;b; Bauer et al., 2023), or transformer-based architecture (Chen & Wang, 2022). The inherent differentiation gives INR the benefit of being independent of resolution, enabling it to depict scenes and objects with outstanding precision and fidelity (Chen et al., 2021; Sajjadi et al., 2022; Jun & Nichol, 2023).

Grattarola & Vandergheynst (2022) proposed a generalized INR, which is the only work dedicated to the study of INR for proteins. The contributions were crucial because generalized INR expanded the use of INR to topological systems that do not possess a well-defined coordinate system. They utilized 2D graph spectral embedding to learn INR for various real-world systems in non-Euclidean domains, including proteins. Nevertheless, although the work demonstrated the capacity to generalize across diverse systems, it necessitated the training of individual Multi-Layer Perceptron (MLP) models for each sample, hence constraining its ability to generalize across datasets. Our study provides evidence that it is feasible to represent protein surfaces using the INR with the Euclidean coordinate system. Furthermore, our study contributes to the area by showcasing the feasibility of a generalizable INR model capable of representing an entire dataset with a single model.

## 3 PRELIMINARIES

### 3.1 PROTEIN GRAPH

Proteins are constructed by 20 different amino acids. Their 3D structures are formed through the chemical bonds and interactions among the atoms of the amino acids and making them naturally suited for graph representation. Based on the GearNet's representation (Zhang et al., 2022), which exhibits high performance for downstream tasks we aim to solve, a protein $\mathcal{P}$ is expressed as a relational graph $\mathcal{G}_\mathcal{P}$, made up of $(\mathcal{V}, \mathcal{E}, \mathcal{R})$. $\mathcal{V}$ is the set of nodes and each node presents a residue in protein and includes the amino acid residue type and 3D coordinate. $\mathcal{E}$ is the set of edges among

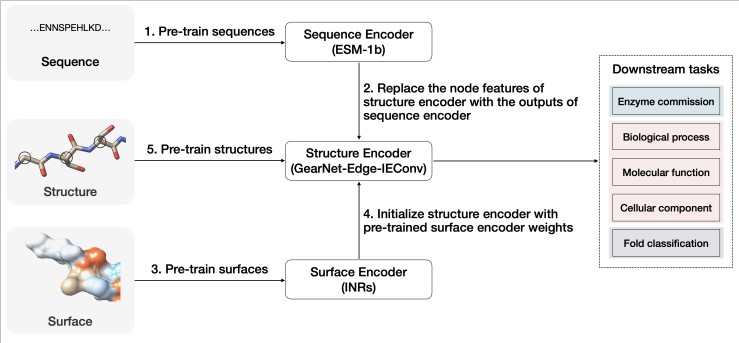

Figure 1: An illustration of our proposed strategy for pre-training sequences, structures, and surfaces to solve downstream tasks.

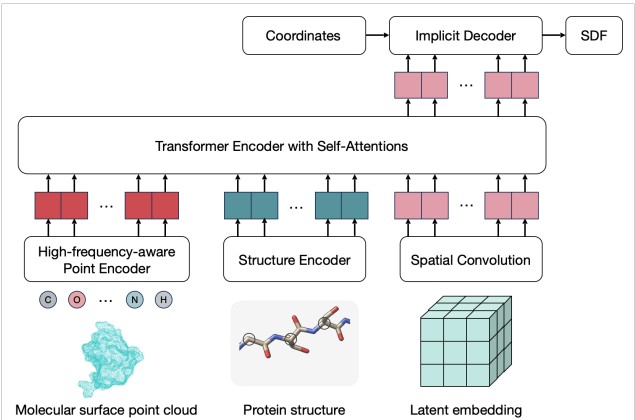

Figure 2: An overview of our ProteinINR architecture. The points tokens, structure tokens, and latent tokens are calculated using high-frequency-aware point encoder, structure encoder (GearNet-Edge-IEConv), and three-dimensional convolution layers, respectively. Points are 16k resolution. Transformer encoders output parameters of an MLP using the tokens, then SDF values are obtained using the parameters for the query coordinates.

nodes with their types $\mathcal{R}$ such as the edges between two residues located within a certain distance on the protein sequence or 3D coordinates.

### 3.2 INRs

To model the surface of the protein, and we utilize Signed Distance Function (SDF) to represent the surface. SDF is a well-established strategy for representing 3D shapes as scalar fields. The SDF is a mathematical expression that assigns a scalar value to a given coordinate $\mathbf{x}$, expressing the distance $d$ between the spatial point and the closest point on the shape's surface as follow:

$$\mathcal{F}(x) = s \colon \mathbf{x} \in \mathbb{R}^3, s \in \mathbb{R}. \tag{1}$$

We employ the methodology of DeepSDF (Park et al., 2019a) and train a model that possess continuous implicit representations, which describe the $\mathcal{F}$ for geometric molecular surfaces. We define the inside surface as $d < 0$ and the outside surface as $d > 0$. Following this definition, the equation $\mathcal{F}(x) = 0$ implies the molecular surface boundary, specifically defining the molecular surface. In summary, we train a model that encode a protein molecular surface and produce INR parameters, which imply the $\mathcal{F}$.

# 4 METHOD

As mentioned earlier, we aim to pre-train sequences, structures, and surfaces of proteins for better protein representation. The sequence, structure, and surface are quite different modalities and thus establishing strategies for pre-training the sequence, structure, and surface is very important. To learn a large volume of structural and sequence data, we employ the "series-fusion" approach which has demonstrated superior performance in previous work (Zhang et al., 2023). First, we pre-train the sequences on sequence encoder and use this encoding as input for the structure encoder. Then, we pre-train the structure encoder on the surfaces using ProteinINR and utilize the weights of pre-trained structure encoder as initial weights for pre-training on the structures. Then, we pre-train the structure encoder on the structure through multi-view contrastive learning based on the approach from Zhang et al. (2022) to obtain the final protein representation. Our pre-training strategy can be seen as continual pre-training (Ke et al., 2022). Finally, we leverage protein representation from the pre-trained model on three modalities to solve downstream tasks. Figure 1 contains an illustration of our pre-training strategy.

## 4.1 GENERALIZABLE IMPLICIT NEURAL REPRESENTATIONS FOR PROTEIN

To effectively pre-train protein surfaces, we employ INRs. In the early stages of INRs, a coordinate-based MLP is trained for each individual instance. However, with the increasing amount of datasets, the computational expense associated with training multiple MLPs for each individual data point has become too costly. Consequently, various solutions have been proposed to develop a generalizable INR to accommodate an entire dataset within a single model. One notable approach, TransINR (Chen & Wang, 2022), entails leveraging Transformer architecture, particularly for INR parameter calculation based on multiple partial views of 3D objects as conditioning inputs. This technique has garnered considerable attention in the field. Building upon these advancements, ProteinINR adopts and first extends these methodologies in the protein field. It represents an expressive and generalizable INR that can effectively capture the shapes of tens of thousands of protein instances within a single model.

### 4.1.1 ENCODING PROTEIN USING POINT AND STRUCTURE ENCODER

The ProteinINR framework first encodes a certain protein instance $\mathcal{P}$ into a protein point set embedding $\mathbf{h}$. ProteinINR inputs the 3D protein asset as a protein point cloud $\mathcal{P} \in \mathbb{R}^{N \times 3}$. The variable $N$ denotes the number of points in the point cloud of the protein molecular surface, and we randomly sampled 16,384 points to input the point encoder in our experiments. ProteinINR utilizes the Dual-scale Point Cloud Recognition (DSPoint) (Zhang et al., 2021) Encoder $\psi$ to address the complex and irregular nature of protein surfaces, which exhibit intricate high-frequency features. This encoder effectively captures a given point cloud's high-frequency and low-frequency characteristics, demonstrating notable efficacy in the tasks that involve high-frequency features, such as point cloud segmentation. Following the process of updating point features through the DSPoint method, we downsample the points into a reduced set of $M$ points $\tilde{\mathcal{P}} \in \mathbb{R}^{M \times 3}$ by utilizing the deformable Kernel Point Convolution (KPConv) networks (Thomas et al., 2019). Ultimately, a learnable linear transformation is implemented on the downsampled points to align the embeddings' hidden dimension prior to cross attention.

RGB values are frequently utilized as characteristics for individual points in point cloud modeling. ProteinINR considers the chemical properties of protein surfaces stemming from their electrical environment as chemical colors. Although MaSIF utilized a pre-computation technique to determine the chemical colors, the computational cost associated with this approach is prohibitively expensive. Fortunately, dMaSIF has shown that it is possible to create a comprehensive representation of chemical properties by utilizing atom category features and distances inside an end-to-end learning framework. Building upon these findings, we adopt a similar approach for protein point cloud chemical color representation. We integrate two essential elements into our approach, namely atom categorical embeddings and the property of Top K closest distances. Incorporating these characteristics into the point cloud encoder leads to the formation of embeddings that encompass the surface's chemical attributes. The utilization of the encoder and chemical features mentioned ensures that ProteinINR represents the protein molecular surface by considering the intricate interaction between the protein's structural and chemical characteristics.

The primary contribution of our study is employing INR training as a pre-training technique to inject the knowledge of protein surface characteristics into the protein structure encoder. In order to accomplish this, we represent an input protein as a structure and incorporate a protein structure encoder into the INR training process, which allows us to encode the protein structure graph $\mathcal{G}_\mathcal{P}$ and generate protein structure embeddings $\mathbf{g} \in \mathbb{R}^{R \times h}$ where $R$ and $h$ are length of residues and length of hidden dimension, respectively. Finally, the embeddings $h$ can be used for various downstream tasks. The utilization of this architectural design enables the protein structure encoder to actively participate in the process of acquiring surface-aware representation learning. As a result, the structure encoder enhances its ability to comprehend and depict protein molecule surfaces comprehensively. We note the extracted point embedding as $\mathbf{p} \in \mathbb{R}^{M \times h}$.

### 4.1.2 SPATIALLY ARRANGED LATENT REPRESENTATIONS

Recently, Spatial Functa (Bauer et al., 2023) has demonstrated improvements in the quality of latent representations when two-dimensional spatial inductive biases are incorporated. Building upon this, we extend the concept to three-dimensional protein surfaces. In ProteinINR, the latent embeddings $\mathbf{z} \in \mathbb{R}^{L \times c}$ with length of $L$ are initially rearranged into a three-dimensional voxel grid $\mathbf{z} \in \mathbb{R}^{i \times j \times k \times c}$, where c, i, j, and k are feature size, width, height, and depth of latent grid respectively. Following that, we implemented 3D convolutions on the reorganized embeddings, which allowed for the incorporation of spatial inductive biases inside the latent embeddings. Finally, latent embeddings are rearranged to have the original shape $\mathbf{z} \in \mathbb{R}^{L \times c}$ and projected through a learnable projection layer to have the feature dimension $\mathbf{z} \in \mathbb{R}^{L \times h}$. While this approach may seem simple, the results are remarkably effective, leading to enhanced INR performance, as further elucidated in our ablation study.

### 4.1.3 TRANSFORMER ENCODER FOR INRS

In ProteinINR, the latent representation (referred to as $\mathbf{z}$) of a protein instance's surface is obtained using a transformer encoder. The initial step is the concatenation of the protein surface point cloud embedding $\mathbf{p}$, structural embedding $\mathbf{g}$, and latent embedding $\mathbf{z}$ as follows:

$$\mathbf{h} = \text{Concat}(\mathbf{p}, \mathbf{s}, \mathbf{z}), \mathbf{h} \in \mathbb{R}^{(M+R+L) \times h} \tag{2}$$

Next, the final latent codes $\mathbf{z}$ are obtained through self-attention processes where protein information is propagated over all protein-related tokens and latent embeddings.

### 4.1.4 INR DECODER AND SDF REGRESSION

In order to strengthen the ability of ProteinINR to capture localized and fine-grained details of local surfaces, we utilize the decoder introduced by (Lee et al., 2023). This decoder has demonstrated a significant improvement of over 50% compared to the prior TransINR model. The improvement is achieved by introducing a locality inductive bias into the INR framework. In ProteinINR, the locality-aware INR decoder $\mathbf{D}_\phi$ utilizes the latent code $\mathbf{z}$ to predict the SDF $\tilde{\mathbf{s}}$ for $K$ query coordinates $\mathbf{x} \in \mathbb{R}^{K \times 3}$ near molecular surface of $N$ protein samples $\mathcal{P}^n$. The optimization of ProteinINR is fulfilled by minimizing the L2 loss between the predicted SDF values and the corresponding SDF values for each SDF sample. Furthermore, clamping techniques are employed to focus the model's attention on the specific details within the vicinity of the surface region. We used the clamp value of 0.2, as employed in the DeepSDF. More detailed steps are followed:

$$\tilde{\mathbf{s}} = \mathbf{D}_\phi(\mathbf{x}, \mathbf{z}) \tag{3}$$

$$\min_{\psi, \mathbf{z}} \frac{1}{NK_n} \sum_{n=1}^{N} \sum_{i=1}^{K_n} \|\text{clamp}(\mathbf{s}, \delta) - \text{clamp}(\tilde{\mathbf{s}}, \delta)\|_2^2 \tag{4}$$

### 4.2 PRE-TRAINING ON SEQUENCES AND STRUCTURES

To effectively learn protein representations from a large volume of structural and sequence data, we employ the "series fusion" approach, which has demonstrated superior performance in previous

work (Zhang et al., 2023). In the "series fusion" architecture, the output from the trained language model is fed into the structure encoder. We utilize ESM-1b (Meier et al., 2021) as the trained language model. To encode a protein graph by learning their structural information, we adopt the GearNet-Edge-IEConv architecture, which performs best across most tasks, as the structure encoder and then we pre-train the structure encoder on structures by employing the multi-view contrastive learning approach (Zhang et al., 2022). The detailed information about the architecture and hyperparameters of structure pre-training we used is described in Appendix A.1.

# 5 EXPERIMENTS AND RESULTS

## 5.1 DATASET PREPARATION FOR PRE-TRAINING

**INR training** Before calculating samples of the signed distance function, we generated the zerolevel surface, namely, the molecular surface, represented by the equation $\mathcal{F}(x) = 0$ in implicit representations. To accomplish this objective, we utilized the MSMS program (Connolly, 1983; Sanner et al., 1996), which is well-established triangulation software for molecular surfaces. Subsequently, we computed the SDF values for the points acquired by the sampling approach utilized in DeepSDF. The sample points are sampled near to the molecular mesh obtained via MSMS. In that case, the SDF values are their distances from the nearest vertices point of a given molecular surface mesh. In this work, 500,000 points were generated for SDF training, serving as the SDF points independent of the protein point cloud input. These points and corresponding SDF values are utilized as the target data for INR training. We train ProteinINR in 50 epochs with learning rate of 1e-4.

**Structure pre-training** To pre-train structural information, we utilize AlphaFold Protein Structure Database version 2 (Varadi et al., 2022) to pre-train the models. We use protein structure prediction data for 20 species and Swiss-Prot (Boeckmann et al., 2003). In-depth details and statistics about the data we used are provided in the Appendix A.2.

## 5.2 EXPERIMENTAL SETTINGS

**Downstream tasks** To quantify representation power of our proposed method, we adopt three downstream tasks. As in GearNet paper, we choose Enzyme Commission (EC) number prediction task and Gene Ontology (GO) term prediction proposed from Gligorijević et al. (2021). Fold Classification (FC) suggested from Hou et al. (2018) is adopted as downstream evaluation as well. EC task is prediction of EC numbers of proteins which represent biomedical reactions they catalyze. GO task is divided into three sub-tasks by their ontologies, biological process (BP), molecular function (MF), cellular component (CC). Each task predicts whether a protein is associated with a specific GO term. For EC and GO tasks, $F_{max}$ and pair-centric area under precision-recall curve (AUPR) values are calculated to measure performance. In the FC task, fold labels of proteins are classified, and mean accuracy is used to evaluate performance.

We evaluate a total of seven models: i) **GearNet**, which is trained directly on the downstream tasks with a structure module; ii) **GearNet-INR**, where the structure module is pre-trained on the surfaces, and then trained on the downstream task; iii) **GearNet-MC**, whose structure module is pre-trained on the structures by multi-view contrastive learning, and then trained on downstream tasks; iv) **GearNet-INR-MC**, whose structure module is pre-trained on the surfaces, subsequently on the structures, and then trained on downstream tasks; v) **ESM-GearNet-MC**, where a sequence encoder is pre-trained, followed by pre-training on the structures; vi) **ESM-GearNet-INR**, where a sequence encoder is pre-trained, followed by pre-training on the surfaces; vii) **ESM-GearNet-INR-MC**, which entails pre-trained a sequence encoder, then pre-training the structure module on the surface, followed by further training on the structure, and finally training on the downstream tasks. We use ESM-1b as the seuqnece encoder and GearNet-Edge-IEConv as the structure encoder. We finetune each task with the datasets as described in Appendix A.3. The model is trained for 50 epochs on EC, 200 epochs on GO, and 300 epochs on fold classification task. We finetune and evaluate the model upon the framework proposed by GearNet (Zhang et al., 2022) and all other settings for finetuning models is same except batch size. We use batch size as 16 per step (8 A100 GPUs and 2 for each GPU) for all experiments.

Table 2: Performance on downstream tasks. We compare the models with and without using the pre-trained weights from ProteinINR. We highlight the cases where performance is the best in terms of $F_{max}$ and AUPR for EC and GO task and mean accuracy for FC in bold. † indicates scores extracted from Xu et al. (2023), which conducted different settings compared to our study.

| Method | EC | | GO-BP | | GO-MF | | GO-CC | | FC | Sum |
|---|---|---|---|---|---|---|---|---|---|---|
| | $F_{max}$ | AUPR | $F_{max}$ | AUPR | $F_{max}$ | AUPR | $F_{max}$ | AUPR | Acc | |
| ESM-1b† | 86.9 | 88.4 | 45.2 | 33.2 | 65.9 | 63.0 | 47.7 | 32.4 | - | - |
| ESM-2† | 87.4 | 88.8 | 47.2 | **34.0** | 66.2 | **64.3** | 47.2 | 35.0 | - | - |
| GearNet | 81.6 | 83.7 | 44.8 | 25.2 | 60.4 | 52.9 | 43.3 | 26.8 | 46.8 | 465.5 |
| GearNet-INR | 81.4 | 83.7 | 44.7 | 26.5 | 59.9 | 52.1 | 43.0 | 27.2 | 47.6 | 466.1 |
| GearNet-MC | 87.2 | 88.9 | 49.9 | 26.4 | 64.6 | 55.8 | 46.9 | 27.1 | 51.5 | 498.3 |
| GearNet-INR-MC | 86.9 | 88.9 | 49.8 | 26.0 | 65.4 | 56.1 | 47.7 | 26.6 | 51.1 | 498.5 |
| ESM-GearNet-MC | 89.0 | 89.7 | **53.5** | 27.5 | **68.7** | 57.9 | 49.4 | 32.4 | **53.8** | 521.9 |
| ESM-GearNet-INR | 89.0 | 90.3 | 50.8 | 33.4 | 67.8 | 62.6 | **50.6** | **36.9** | 48.9 | **530.3** |
| ESM-GearNet-INR-MC | **89.6** | **90.3** | 51.8 | 33.2 | 68.3 | 58.0 | 50.4 | 35.7 | 50.8 | 528.1 |

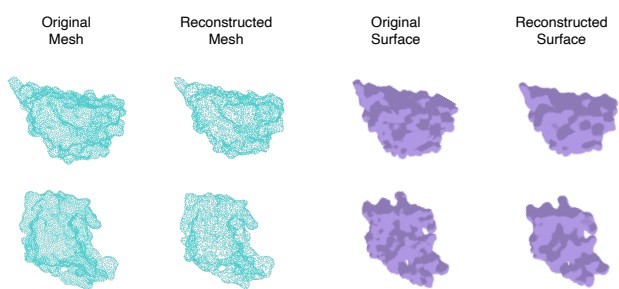

Figure 3: Above images are the examples of reconstructed meshes and surfaces from ProteinINR for given proteins. ProteinINR preserves the intricate details of irregular protein surfaces, particularly capturing features such as ring-like and hole shapes with remarkable fidelity.

## 5.3 EXPERIMENTAL RESULTS

**Representing protein surface shapes using ProteinINR** The procedure for acquiring a triangular mesh that corresponds to a specific protein using INR parameters from ProteinINR is outlined as follows. Initially, the SDFs are calculated for the vertices of a voxel grid with a regular size of 128. Following this, the marching cubes algorithm (Chernyaev, 1995) is employed to compute the mesh. Protein surface samples reconstructed using ProteinINR are depicted in Figure 3. It is worth mentioning that the protein molecular surfaces exhibit significant irregularity and possess high-frequency properties. Intriguingly, ProteinINR effectively preserves intricate information, even hole or ring-like shapes. In addition, we calculated the Chamfer distance between the ground truth and the reconstructed meshes for the test set. A subset of 30,000 data points was selected, and the computed average Chamfer distance A.4 was 0.003. The number might be quite decent in the context of Chamfer distances for natural 3D objects as reported in studies (Mescheder et al., 2019; Park et al., 2019b; Sitzmann et al., 2020a; Liu et al., 2023) related to SDF reconstruction. These findings indicate that ProteinINR effectively acquires generalizable INRs that can accurately depict the uneven surfaces of proteins.

**Downstream evaluation** We compare the performance of the structure encoder initialized with weights from a pre-trained ProteinINR model and without such initialization across various downstream tasks related to protein function. Intriguingly, we can see that **ESM-GearNet-INR-MC** and **ESM-GearNet-INR** outperform the previous state-of-the-art model, **ESM-GearNet-MC**, when taking the summation of all scores. This demonstrates our main contribution clearly, emphasizing that incorporating surface-related features, which have not been explored by previous models, into protein pre-training representation learning enables comprehensive representation learning for proteins. Additionally, we observe a rapid decrease in pre-training loss as depicted in Figure 4, which provides an additional evidence.

Table 3: Results on EC task depending on pre-training order.

| Method | Pre-training order | EC | | GO-BP | | GO-MF | | GO-CC | | FC | Sum |
|---|---|---|---|---|---|---|---|---|---|---|---|
| | | $F_{max}$ | AUPR | $F_{max}$ | AUPR | $F_{max}$ | AUPR | $F_{max}$ | AUPR | Acc | |
| **ESM-GearNet-INR-MC** | sequences → surfaces → 3D-structures | 89.6 | 90.3 | 51.8 | 33.2 | 68.3 | 58.0 | 50.4 | 35.7 | 50.8 | 528.1 |
| **GearNet-INR-MC** | surfaces → 3D-structures | 86.9 | 88.9 | 49.8 | 26.0 | 65.4 | 56.1 | 47.7 | 26.6 | 51.1 | 498.5 |
| **GearNet-MC-INR** | 3D-structures → surfaces | 84.1 | 86.0 | 46.9 | 25.9 | 62.1 | 54.3 | 44.8 | 27.2 | 47.6 | 478.9 |

Protein function primarily occurs on the surface and is closely associated with surface features. The observed enhancement in protein function tasks indicates that acquiring surface understanding using INR is advantageous. In contrast, a noticeable enhancement in performance is not observed in the FC task. Since surface features imply higher representations of the outer part of protein structure, these features may not contribute substantially to classifying the overall fold structure. Otherwise, as the process of pre-training progresses, the loss gap between models diminishes. We attribute this trend to the nature of our encoder, which focuses solely on alpha carbons. While surface information is derived from full-atom information, the encoder only learns from alpha carbon structures. So, we hypothesize that the 2nd stage mutual information maximization during pre-training on structure data biases the model toward alpha carbon structure information after 1st stage surface pre-training.

Nonetheless, even under these limited conditions, the models including the protein surface modality show performance gains.

**Experiment on order** Considering the original results (Table 2), it is evident that ESM has the most dominant impact on the downstream tasks, revealing the supportive role of structure and surface in enhancing performance on downstream tasks. Table 3 enables us to compare the significance of structure and surface pre-training while excluding the dominant influence of sequence. We can see that the structure encoder (GearNet-INR-MC), which learned structure information last, had superior performance compared to GearNet-MC-INR, which was pre-trained in the opposite manner. Based on the results of GearNet-INR (466.1) and GearNet-MC (498.3) shown in Table 2, it seems that in the absence of pre-training on sequences, structure pre-training has a greater influence on downstream tasks compared to surfaces. We conjecture that this observation supports the findings shown in Table 3.

**Ablation study on 3D latent embedding** In ProteinINR, we incorporate a 3D convolution layer to introduce a spatial inductive bias to the latent space. To evaluate the effect of the approach, we conduct an analysis of the learning curve of INR when incorporating or excluding spatial inductive bias. As depicted in Figure 5 in Appendix, the existence of spatial inductive bias clearly enhances the learning of INR.

# 6 CONCLUSION

We propose a pre-training strategy for learning from sequences, structures, and surfaces of proteins to achieve better protein representation. For the first time, we use INR to pre-train the protein surface, introducing a method we call ProteinINR. We confirm that ProteinINR effectively reconstructs the protein surfaces. Moreover, the results on the downstream tasks demonstrate that learning the protein surface can lead to better protein representation.

Our work represents an important step towards incorporating protein surfaces, which play a crucial role in protein functions. There are several interesting avenues for further research: generating new proteins from the latent representation of surfaces we pre-train; applying our approach to other types of molecules, such as small-molecule drugs; and identifying a better strategy for integrating all three modalities, particularly the effective integration of surface and structure pre-training.

Meanwhile, our approach has a limitation of dependency on protein structure, so the use of predicted structures may worsen the performance of our method for proteins without known structures.

## 7 REPRODUCIBILITY

The architectural design of ProteinINR is influenced by TransINR, and we utilize the decoder model introduced by Lee et al (Lee et al., 2023). The DSPoint and KPConv, and GearNet are implemented from their official codes. The training dataset used in pre-training structures is prepared from the AF2 prediction dataset, similar to in GearNet. The procedure for generating SDF data is implemented in accordance with the approach described in the DeepSDF framework. To assess the performance of downstream tasks, we use the well-published TorchDrug framework (Zhu et al., 2022). We describe detailed information regarding the training and evaluation in Section 4, Section 5, and Appendix.

## 8 ETHICAL STATEMENT

In this work, we focus on advancing the topic of protein representation learning by incorporating surface information alongside sequence and 3D structure-based representations. We acknowledge the importance of ethical considerations in scientific research and we aim to provide further clarification on the following ethical aspects.

We provide transparent and comprehensive details about our methodology, experiments, and results. We list any limitations or potential biases in our research.

Our research aims to elucidate the significance of surfaces in protein representation learning, hence potentially influencing drug discovery and enzyme development. This would have the potential impact on a wide range of applications, including the identification of innovative therapeutic targets, the development of more promising drugs, improvements in agricultural productivity, and ultimately, improvements in human health.

Our purpose of this research is to make a constructive and positive contribution to the domain of protein science while upholding the ethical conduct of our research.

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

# A APPENDIX

## A.1 DETAILED INFORMATION ON PRE-TRAINING

### A.1.1 GEARNET-EDGE-IECONV FOR STRUCTURE ENCODER

To encode the structural information of proteins, we use the GearNet-Edge-IEConv architecture (Zhang et al., 2022). In GearNet architecture, a protein is represented by a graph $\mathcal{G} = (\mathcal{V}, \mathcal{E}, \mathcal{R})$ where $\mathcal{V}$ represents residues of proteins, $\mathcal{E}$ denotes edges between residues, and $\mathcal{R}$ represents edge types. Edges in GearNet are of three types: sequential edges, which are edges between two residues located within a certain distance on the protein sequence; radius edges, which are edges between two residues that have a Euclidean distance of less than a specific value in 3D coordinates; and k-nearest neighbors edges, which are edges between a specific node and its k-nearest neighbors in terms of Euclidean distance in 3D coordinates. GearNet-Edge-IEConv has two additional elements compared to GearNet: i) the Edge element, which transforms edges into nodes resulting in $\mathcal{G}' = (\mathcal{V}', \mathcal{E}', \mathcal{R}')$ and facilitates message passing between edges, and ii) the IEConv element, which applies a learnable kernel function to the edge, inspired by the previous work (Hermosilla et al., 2020).

### A.1.2 PRE-TRAINING OF PROTEIN STRUCTURES

To pre-train GearNet-Edge-IEConv, we employ the multi-view contrastive learning approach, which shows the highest performance in the previous work (Zhang et al., 2022). Multi-view contrastive learning aims to make the embeddings of related substructures similar, while rendering the embeddings of unrelated substructures distinct, akin to SimCLR (Chen et al., 2020). Substructures are extracted either by taking a subsequence from the sequence or by extracting a subspace in 3D space. Table 4 presents the hyperparameters used in pre-training of structural data for ESM-GearNet-IEConv and GearNet-IEConv. We save the model checkpoints every 5 epochs, and among the saved checkpoints, we use the checkpoint with the lowest loss for downstream tasks. We use 64 NVIDIA A100 80GB gpus for pre-training.

Table 4: GearNet-Edge-IEConv and ESM-GearNet-Edge-IEConv hyperparameters

|  | GearNet-Edge-IEConv | ESM-GearNet-Edge-IEConv |
|---|---|---|
| **Subsequence maximum length** | 50 | 50 |
| **Subspace minimum neighbors** | 15 | Not used |
| **Sequence model** | Not used | ESM-1b |
| **Batch size** | 48 | 48 |
| **Optimizer** | Adam | Adam |
| **Learning rate** | 1.0e-3 | 2.0e-4 |
| **# epochs** | 50 | 50 |

## A.2 DATASETS USED IN STRUCTURE PRE-TRAINING

We utilize protein structure prediction data for 20 species and Swiss-Prot (Boeckmann et al., 2003) from AlphaFold Protein Structure Database version 2 (Varadi et al., 2022). After processing to convert pdb files to protein graphs and remove the proteins with errors, the statistics of the graphs we finally used are presented in Table 5.

## A.3 DETAILED INFORMATION ON DOWNSTREAM TASK

For downstream task, we basically use the framework provided by GearNet(Zhang et al., 2022) and all experiments shown in Table 2 are performed under same conditions. The number of datasets used in each downstream task is described in Table 6. We finetune the models with batch size as 16 per step. All other settings are used as provided by the framework. The models demonstrating the highest performance on validation set are chosen to report the results in Table 2.

Table 5: The number of protein structures used per species

| Proteome ID | Taxonomy | # structures |
|---|---|---|
| UP000006548 | Arabidopsis thaliana | 27393 |
| UP000001940 | Caenorhabditis elegans | 19658 |
| UP000000559 | Candida albicans | 5956 |
| UP000000437 | Danio rerio | 24595 |
| UP000002195 | Dictyostelium discoideum | 12592 |
| UP000000803 | Drosophila melanogaster | 13424 |
| UP000000625 | Escherichia coli | 4363 |
| UP000008827 | Glycine max | 55747 |
| UP000005640 | Homo sapiens | 23280 |
| UP000008153 | Leishmania infantum | 7903 |
| UP000000805 | Methanocaldococcus jannaschii | 1772 |
| UP000000589 | Mus musculus | 21571 |
| UP000001584 | Mycobacterium tuberculosis | 3988 |
| UP000059680 | Oryza sativa subsp. japonica | 43623 |
| UP000001450 | Plasmodium falciparum | 5162 |
| UP000002494 | Rattus norvegicus | 21209 |
| UP000002311 | Saccharomyces cerevisiae | 6026 |
| UP000002485 | Schizosaccharomyces pombe | 5123 |
| UP000008816 | Staphylococcus aureus | 2885 |
| UP000002296 | Trypanosoma cruzi | 18992 |
| UP000007305 | Zea mays | 39258 |
| Swiss-Prot | - | 541938 |
| **Total** | - | 906458 |

Table 6: The number of datasets for downstream tasks.

| Dataset | # Train | # Validation | # Test |
|---|---|---|---|
| **Enzyme Commission** | 15,170 | 1,686 | 1,860 |
| **Gene Ontology** | 28,305 | 3,139 | 3,148 |
| **Fold Classification** | 12,312 | 736 | 718 |

## A.4 CHAMFER DISTANCE

The Chamfer distance is a commonly used metric for shape evaluation, favored for its simplicity Fan et al. (2017). When comparing two point sets, S1 and S2, this metric is the summation of nearest-neighbor distances between each point and its closest corresponding point in the other set as follows:

$$d_{\text{Chamfer distance}}(S_1, S_2) = \sum_{x \in S_1} \min_{y \in S_2} \|x - y\|_2^2 + \sum_{y \in S_2} \min_{x \in S_1} \|x - y\|_2^2 \qquad (5)$$

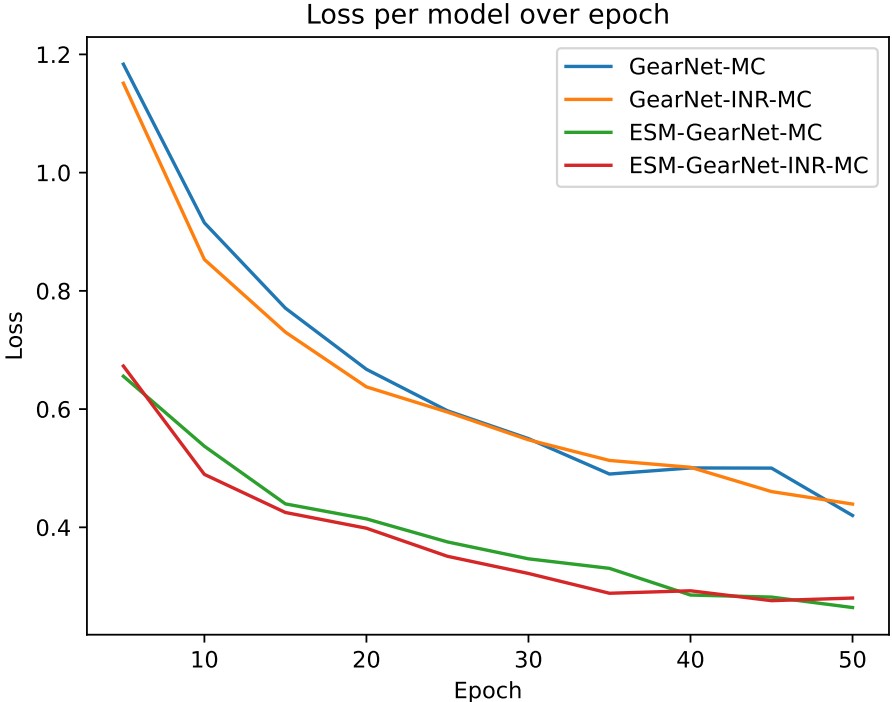

Figure 4: The loss of multi-view contrastive learning of GearNet-MC, GearNet-INR-MC, ESM-GearNet-MC, and ESM-GearNet-INR-MC. Compared to GearNet-MC and ESM-GearNet-MC, we observe that the GearNet-INR-MC and ESM-GearNet-INR-MC, which learn surface information, exhibit a faster decrease in loss initially.

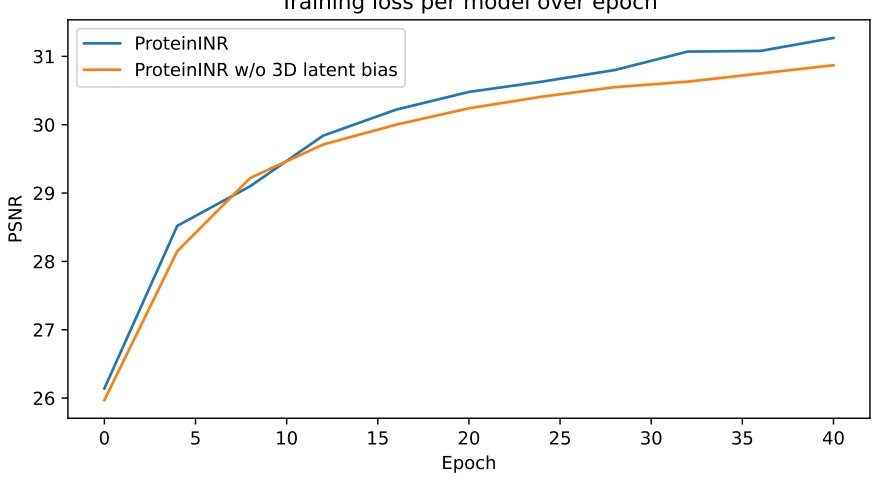

Figure 5: Training curves of ProteinINR with regard to the utilization of 3D latent bias.