# OpenReview forum: "Pre-training Sequence, Structure, and Surface Features for Comprehensive Protein Representation Learning"
_ICLR.cc/2024/Conference — ICLR 2024 poster_

### Official Review · Reviewer_rs2T · 2023-10-26

**Soundness:** 2 fair
**Presentation:** 2 fair
**Contribution:** 3 good
**Rating:** 5
**Confidence:** 5

**Summary:**

This paper proposes to jointly pretrain protein surface and structure based on the pretrained ESM-1b. Consequently, the final protein representation includes three-kind information, which are protein sequence, surface, and structure. The pretrained protein representations are finetuned on three downstream tasks, which are enzyme classification, gene ontology term prediction and fold classification. The propose model performs best in most cases among all the ablation models.

**Strengths:**

As far as I know, this paper is the first one to involve surface information into protein pretraining, and then applied the pretrained protein representations for downstream protein understanding tasks.

**Weaknesses:**

1. **Many of the components used in the proposed method are not new.** For example, for surface preparation, the paper just followed the apporoach proposed in [1]. For surface dowmsampling, the paper use DSPoint method.  For latent representation learning, the paper follow Spatial Functa. It seems the proposed method combine several existing modules to get a new pipeline.

[1] Fast end-to-end learning on protein surfaces. CVPR 2021.

2. **The paper lacks some important baselines.** In the main results, the paper only compares to the ablation models, which seems not enough. At least, the paper should compare to [1], [2]. Besides, the paper seems lacks the baseline of directly finetuning ESM-1b (ESM-2). I don't know how much benefits  that adding the additional surface and structure pretraining will bring to directly finetuning ESM series.

[2] LEARNING HARMONIC MOLECULAR REPRESENTATIONS ON RIEMANNIAN MANIFOLD. ICLR 2023.

3. **Some claims are not right.** In the first several lines of page 6, the paper mentioned "Incorporating these characteristics into the point cloud encoder leads to the formation of embeddings that encompass both the surface’s geometric structure and chemical attributes." On the surface, the author only used atom category and distance, which are chemical features. For geometric features, the author may need to calculate Gaussian curvature, mean curvature, heat kernel signature and something like that, which are geometric features. Besides, I think the calim "pre-training sequence, structure, and surface features" might be not true because there is no sequence pretraining stage involved. Instead, this paper used pretrained ESM-1b, and continued pretraining based on this model, which is continual learning.

4. **The paper missed some important details.**  For example, in SDF, we need the atom type for the current vertex and its neighbors. However, the author didn't mention what atoms they used. The four atoms on backbone? Or the full 44-kind atoms as in AlphaFold2?

5. **Function evaluation:** It's just a suggestion. One of the most important motivation of this paper is: surface encodes the function of proteins, so involving surface can get better protein representation. Therefore, it's very natural to apply the learned protein representations to prediction the protein functions. However, the paper didn't do such kind of experiments. I suggest the author add some protein function evaluation, such as protein fitness landscape.

**Questions:**

I also have the following questions:

1. In 4.1.1, N= 16384. What does N refers to? Residue number? R refers to this. Number of vertices on surface? But in 5.1, the author mentioned they sampled 500,000 points for each protein. Besides, Does 500,000 sample points refer to the point cloud after downsampling? According to my opinion, using MSMS, there would be 100-120 vertices around each residue, which means 500,000 points at least represents a protein with length 5,000. That is a very long sequence. I don't believe the average minimum length of the pretraining proteins is 5,000.

2. In Figure1, there is surface encoder and sequence encoder. In title 4.1.1, there is protein encoder. In title 4.1.3, there is Transformer encoder. It's kind of confusing. The author may need to unify the name and give an overall description.

3. Why the author used ESM-1b instead of ESM-2. ESM-2 performs better.

4. In Figure 3, is the reconstructed mesh obtained through the donwsampled vertices? And the original one is the one generated by MSMS?

---

> ### Author Response · Authors · 2023-11-20
> **Thank you. Our updates. [1/2]**
>
> We thank the reviewer for the valuable feedback that helps us to improve our work. Please see our responses below:
>
> > W1: Many of the components used in the proposed method are not new. For example, for surface preparation, the paper just followed the apporoach proposed in [1]. For surface dowmsampling, the paper use DSPoint method. For latent representation learning, the paper follow Spatial Functa. It seems the proposed method combine several existing modules to get a new pipeline.
>
> **Response**
>
> We agreed that we utilized diverse existing approaches to develop generalizable protein implicit neural representations. Nevertheless, this our endeavor resulted in the first development a generalizable implicit neural representation for protein surfaces. Furthermore, utilizing the implicit neural representation for pre-training on protein representations is a unique approach and this also confirms that our contribution is original in this venue.
>
> ---
>
> > W2: The paper lacks some important baselines. In the main results, the paper only compares to the ablation models, which seems not enough. At least, the paper should compare to [1], [2]. Besides, the paper seems lacks the baseline of directly finetuning ESM-1b (ESM-2). I don't know how much benefits that adding the additional surface and structure pretraining will bring to directly finetuning ESM series.
>
> **Response**
>
> We appreciate your constructive feedback. While our approach is similar to the previous works [1] and [2] in terms of studying protein surfaces, our main contribution which added the surface to pretraining for better protein representation is quite different the contribution of and [1] and [2]. The contribution of our framework is to suggest pre-training strategy that inject surface understanding to structure encoder while that of [1] and [2] is to suggest a protein encoder that process surface features in there proposed way. Applying [1] and [2] to protein encoder in our ProteinINR could be helpful and open up another research avenue. Also, we added sentences about [2] paper in related works. Additionally, we have included the both ESM-2 as a baseline in response to your suggestion and ESM-GearNet-INR to show the impact of surface pretraining.
>
>
> - [1] Fast end-to-end learning on protein surfaces. CVPR 2021
> - [2] LEARNING HARMONIC MOLECULAR REPRESENTATIONS ON RIEMANNIAN MANIFOLD. ICLR 2023
>
> ---
>
> > W3: Some claims are not right. In the first several lines of page 6, the paper mentioned "Incorporating these characteristics into the point cloud encoder leads to the formation of embeddings that encompass both the surface’s geometric structure and chemical attributes." On the surface, the author only used atom category and distance, which are chemical features. For geometric features, the author may need to calculate Gaussian curvature, mean curvature, heat kernel signature and something like that, which are geometric features. Besides, I think the calim "pre-training sequence, structure, and surface features" might be not true because there is no sequence pretraining stage involved. Instead, this paper used pretrained ESM-1b, and continued pretraining based on this model, which is continual learning.
>
> **Response**
>
> We thank to the valuable comment. As you pointed out, the phrase "incorporating chemical features" is indeed accurate, and we made the corresponding modifications in the manuscript. Moreover, it is correct that our approach takes the form of continual pre-training (https://arxiv.org/abs/2302.03241), so we have included a comment in the publication to address this.
>
> ---
>
> > W4: The paper missed some important details. For example, in SDF, we need the atom type for the current vertex and its neighbors. However, the author didn't mention what atoms they used. The four atoms on backbone? Or the full 44-kind atoms as in AlphaFold2?
>
> **Response**
>
> Thank you for your valuable comment. We have added a sentence addressing the use of full atoms in calculating SDF for the molecular mesh obtained using MSMS.
>
> ---

---

> ### Author Response · Authors · 2023-11-20
> **Thank you. Our updates. [2/2]**
>
> We thank the reviewer for the valuable feedback that helps us to improve our work. Please see our responses below:
>
> > W5: Function evaluation: It's just a suggestion. One of the most important motivation of this paper is: surface encodes the function of proteins, so involving surface can get better protein representation. Therefore, it's very natural to apply the learned protein representations to prediction the protein functions. However, the paper didn't do such kind of experiments. I suggest the author add some protein function evaluation, such as protein fitness landscape.
>
> **Response**
>
> We appreciate your constructive feedback. We think predicting EC and GO (molecular function, biological process, cellular component) annotations is a task of related to predicting protein functions, and we already have results for these tasks. The protein fitness landscape prediction is a valuable suggestion. To explore this, we can utilize FLIP(https://github.com/J-SNACKKB/FLIP) or PEER (https://github.com/DeepGraphLearning/PEER_Benchmark) benchmarks; however, the benchmarks primarily rely on sequence information. Since our methodology necessitates the use of protein structures, structure prediction would be necessary in cases where proteins do not have existing structures in the benchmarks. Conducting this during the rebuttal period may be challenging, and the accuracy of the predicted structures could influence the performance of downstream tasks. We appreciate the insightful suggestion, and it seems like a worthwhile endeavor for future exploration.
>
> ---
>
> > Q1: In 4.1.1, N= 16384. What does N refers to? Residue number? R refers to this. Number of vertices on surface? But in 5.1, the author mentioned they sampled 500,000 points for each protein. Besides, Does 500,000 sample points refer to the point cloud after downsampling? According to my opinion, using MSMS, there would be 100-120 vertices around each residue, which means 500,000 points at least represents a protein with length 5,000. That is a very long sequence. I don't believe the average minimum length of the pretraining proteins is 5,000.
>
> **Response**
>
> The variable "N" denotes the number of points in the point cloud of the protein molecular surface, and we randomly sampled 16,384 points to input the point encoder in our experiments. Additionally, 500,000 points were generated for SDF training, serving as the SDF points independent of the protein point cloud input. These points and corresponding SDF values are utilized as the target data for INR training.
>
> ---
>
> > Q2: In Figure1, there is surface encoder and sequence encoder. In title 4.1.1, there is protein encoder. In title 4.1.3, there is Transformer encoder. It's kind of confusing. The author may need to unify the name and give an overall description.
>
> **Response**
>
> Thanks for this helpful comment. We updated the manuscript.
>
> ---
>
> > Q3: Why the author used ESM-1b instead of ESM-2. ESM-2 performs better.
>
> **Response**
>
> At the beginning of our research, we employed ESM-1b, more precisely ESM-GearNet-MC, as our baseline. We kept this setup throughout the all experiments to ensure consistency. (Reference: https://openreview.net/forum?id=AAML7ivghpY)
> We acknowledge the advice and also think that utilizing ESM-2 may result in enhanced performance. However, considering the time limitations of rebuttal, we intend to evaluate it in future endeavors. In addition, we have included performance for ESM-2 from the cited literature in our baseline for comparison.
>
> ---
>
> > Q4: In Figure 3, is the reconstructed mesh obtained through the donwsampled vertices? And the original one is the one generated by MSMS?
>
> **Response**
>
> The original mesh was obtained from MSMS, while the reconstructed mesh was calculated by computing SDF on grid coordinates and subsequently using marching cubes.
>
> ---

---

> > ### Comment · Reviewer_rs2T · 2023-11-22
> > **Reply to the rebuttal**
> >
> > I appreciate the efforts the authors have put into the rebuttal. Some of my concerns have been addressed. I have a follow-up question: It seems that finetuning ESM2 did achieve better results on two of the four tasks on AUPR. Did that demonstrate continue learning based on ESM maybe cannot perform better than direct finetuning?

---

> ### Author Response · Authors · 2023-11-22
> **Thank you. Our updates for the reply.**
>
> We thank the reviewer for the valuable feedback that helps us to improve our work. Please see our responses below:
>
> > C1: I appreciate the efforts the authors have put into the rebuttal. Some of my concerns have been addressed. I have a follow-up question: It seems that finetuning ESM2 did achieve better results on two of the four tasks on AUPR. Did that demonstrate continue learning based on ESM maybe cannot perform better than direct finetuning?
>
> **Response**
>
>
> We thank you for the comment. Although ESM-2 performs better regarding the two downstream tasks, our approach is more effective regarding the total summation except FC task (see the table below). These results indicate that including surface and structural modalities during pre-training can result in better representations than fine-tuning the ESM model, which is trained only on the sequence modality. Furthermore, similar findings are reported in the recent ESM-GearNet paper (https://arxiv.org/pdf/2303.06275.pdf, with slightly different experimental settings from ours, e.g., batch size).
>
>
>
> |Model|Sum|
> |---|---:|
> |ESM-GearNet-INR-MC|477.3|
> |ESM-GearNet-INR|481.4|
> |ESM-GearNet-MC|468.1|
> | GearNet-INR-MC   |  447.4  |
> | GearNet-MC-INR   |  431.3  |
> |  GearNet-MC  |  446.8  |
> |  GearNet-INR  |  418.5  |
> |  GearNet    |   418.7  |
> |  ESM-2   |  470.1  |
> | ESM-1b    |  462.7  |

---

### Official Review · Reviewer_VAn9 · 2023-10-28

**Soundness:** 3 good
**Presentation:** 3 good
**Contribution:** 3 good
**Rating:** 6
**Confidence:** 3

**Summary:**

While there are numerous works on pre-training protein representations, most studies have focused on sequences-only, structures-only, or a hybrid of sequences and structures. This work proposes to further incorporate protein surface information for the pre-training. It employed a series-fusion approach where a model is pre-trained in the order of sequences, surfaces, and structures. Another essential part of the work is in the surface encoder which learns Implicit Neural Representations. The surface encoder is based on a proposed ProteinINR framework which consists of a transformer encoder on top of a point encoder, a structure encoder, and spatial latent representations. The experiments demonstrate that ProteinINR can reconstruct protein surfaces pretty well and the pre-trained representations outperform previous SOTA on several downstream tasks.

**Strengths:**

According to the authors’ claim, this is the first work to propose a pre-training scheme for protein representation that incorporates sequence, structures, and especially surfaces. The authors adopted several ideas from previous works to develop ProteinINR which can effectively learn surface characteristics of proteins. I think they mostly explained the proposed method clearly with appropriate background information and without overstating their contributions. They demonstrate the proposed method is effective in learning better protein representations. The significance of the proposed method could vary, but I believe it can help a broad range of researchers with proper further actions.

**Weaknesses:**

- [Reproducibility] Although the authors provided implementation details, they didn’t have any statements regarding public availability. Since the point of pretrained representations is in using them for various downstream tasks, I think that their significance becomes quite small without making ways to obtain the pretrained representations publicly available.
- [Ablations] Would it be possible to develop a surface encoder without implicit neural representations? While the authors discussed the advantage of INR compared to previous methods to represent surfaces, current experiments do not show how important INR is compared to them.
- [Experiments] One of the clear weaknesses of the proposed approach is that it eventually relies on pretraining the structure encoder, GearNet. In other words, it's only applicable for downstream tasks where protein structures are known, which might be the most important in a real world. I'm curious whether authors have thought about downstream tasks where only sequences are available and structures are not. It would be interesting to see how the performance of pre-trained models changes with the usage of predicted protein structures (maybe from AlphaFold) instead of the true ones.

Minor comments
- [Introduction] It would be nice to have more explanation on protein surface characteristics, particularly regarding the relationship between structural features and molecular surfaces. Unfamiliar readers might think that surface information is already well-contained within 3D structures, such that cannot clearly understand the need for incorporating surface information for the pre-training.
- [Figure 2] xyz seems to indicate coordinates, but it's inconsistent with notations from the texts.
- [Sec 4.1.1.] Does the protein encoder mean point encoder? Due to the prevalent terminologies (point encoder, structure encoder, protein encoder, etc.), there seems to be a little confusing use of terminologies. The authors might want to take a look into the issue.
- [Experiments] Can you further provide the performance of ESM-GearNet-INR? It would help to show the effectiveness of INR with sequence pretraining but without structure pretraining.
- [Sec 5.3] It would be better to explain the Chamfer distance.

**Questions:**

- [Reproducibility] Do you have plans for making pre-trained representations and both training/evaluation codes publicly available?
- [Ablations] Can you show how INR is important in the pre-training framework compared to previous methods to represent 3D surfaces?
- [Experiments] Have the authors checked the possible inclusion of data from downstream tasks within the pre-training data?
- [Experiments] Can the pre-trained representations be helpful for downstream tasks where only sequences are available and structures are not?

---

> ### Author Response · Authors · 2023-11-20
> **Thank you. Our updates. [1/2]**
>
> We thank the reviewer for the valuable feedback that helps us to improve our work. Please see our responses below:
>
> > W3: [Experiments] One of the clear weaknesses of the proposed approach is that it eventually relies on pretraining the structure encoder, GearNet. In other words, it's only applicable for downstream tasks where protein structures are known, which might be the most important in a real world. I'm curious whether authors have thought about downstream tasks where only sequences are available and structures are not. It would be interesting to see how the performance of pre-trained models changes with the usage of predicted protein structures (maybe from AlphaFold) instead of the true ones.
>
> **Response**
>
> We appreciate the constructive feedback. While we believe it's a promising topic, the structure and the surface features in our methodology are determined by the structure. Therefore, if the predicted structure is inaccurate, it could have a substantial impact on the performance and efficacy of our approach. We think that experimenting with datasets consisting of predicted structures in the future could lead to intriguing analyses.
>
> ---
>
> > C1: [Introduction] It would be nice to have more explanation on protein surface characteristics, particularly regarding the relationship between structural features and molecular surfaces. Unfamiliar readers might think that surface information is already well-contained within 3D structures, such that cannot clearly understand the need for incorporating surface information for the pre-training.
>
> **Response**
>
>  Thank you for your valuable feedback. We added the following paragraph to the introduction.
> "The protein structure can be divided into the atoms comprising the backbone and the side chains. In this context, the protein surfaces are determined by both backbone and side chain atoms. However, traditional protein structure encoders typically process protein 2D graphs or 3D geometric graphs that only contain alpha carbon or backbone atoms, respectively. Consequently, state-of-the-art representations often lack consideration for side-chain information."
>
> ---
>
> > C2: [Figure 2] xyz seems to indicate coordinates, but it's inconsistent with notations from the texts.
>
> **Response**
>
> Thanks for your comments. We fixed it and updated the manuscript.
>
> ---
>
> > C3: [Sec 4.1.1.] Does the protein encoder mean point encoder? Due to the prevalent terminologies (point encoder, structure encoder, protein encoder, etc.), there seems to be a little confusing use of terminologies. The authors might want to take a look into the issue.
>
> **Response**
>
> Thanks for your comments. We fixed it and updated the manuscript.
>
> ---
>
> > C4: [Experiments] Can you further provide the performance of ESM-GearNet-INR? It would help to show the effectiveness of INR with sequence pretraining but without structure pretraining.
>
> **Response**
>
> Thanks for your productive comment. We have conducted the experiments in accordance with the remark, and the results are presented in the table 3 (ESM-GearNet-INR). Interestingly, when comparing with ESM-GearNet-INR-MC, ESM-GearNet-INR exhibited higher performance in several tasks and demonstrated the highest performance on average. When we conducted an experiment where surface pre-training was performed without ESM, we found that the performance improvement was relatively modest compared to structure pre-training alone, so we initially excluded this experiment. However, following your comment, we found that the integration of surface pre-training with ESM resulted in noteworthy enhancements in performance. This result serves as evidence for the importance of the surface modality, and we have added these findings to the manuscript. Thanks for the valuable feedback; it has enriched the content of the paper.
>
> ---
>
> > C5: [Sec 5.3] It would be better to explain the Chamfer distance.
>
> **Response**
>
> Thanks for your productive comment. We added the sentences about chamfer distance in Appendix A.4
>
> ---
>
> > W1: [Reproducibility] Although the authors provided implementation details, they didn’t have any statements regarding public availability. Since the point of pretrained representations is in using them for various downstream tasks, I think that their significance becomes quite small without making ways to obtain the pretrained representations publicly available.
> >
> > Q1 [Reproducibility] Do you have plans for making pre-trained representations and both training/evaluation codes publicly available?
>
> **Response**
>
> We are planning to make our model publicly available.
>
> ---

---

> ### Author Response · Authors · 2023-11-20
> **Thank you. Our updates. [2/2]**
>
> > W2: [Ablations] Would it be possible to develop a surface encoder without implicit neural representations? While the authors discussed the advantage of INR compared to previous methods to represent surfaces, current experiments do not show how important INR is compared to them.
> >
> > Q2: [Ablations] Can you show how INR is important in the pre-training framework compared to previous methods to represent 3D surfaces?
>
> **Response**
>
> We appreciate your constructive review. We utilized INR to represent continuous surfaces, which is the crucial property of protein surfaces. Your suggestion to explore alternative methods for 3D surface representation is a valuable idea and could serve as another promising avenue for future work. For instance, representing the surface as a point cloud could allow for incorporating masked point modeling into pre-training.
>
> ---
>
>
> > Q3: [Experiments] Have the authors checked the possible inclusion of data from downstream tasks within the pre-training data?
>
> **Response**
>
> We appreciate the insightful comment. We primarily utilized the GearNet codebase (https://github.com/DeepGraphLearning/GearNet) to download and use pretraining and downstream task data. We did not perform additional checks for inclusion of data from downstream task within the pretraining.
>
> ---
>
> > Q4: [Experiments] Can the pre-trained representations be helpful for downstream tasks where only sequences are available and structures are not?
>
>
> **Response**
>
> We appreciate the constructive feedback. Pre-trained representations could indeed be beneficial. However, the structure and the surface features in our methodology are determined by the structure. Therefore, if the predicted structure is inaccurate, it could have a substantial impact on the performance and efficacy of our approach. We think that experimenting with datasets consisting of predicted structures in the future could lead to intriguing analyses.
>
> ---

---

> > ### Comment · Reviewer_VAn9 · 2023-11-21
> > **Post-rebuttal comments**
> >
> > I appreciate the authors' detailed responses. They have addressed most of my comments about the paper, and I'm also delighted to read the revised paper. Here are a couple of follow-up questions:
> > - Considering that ESM-GearNet-INR outperforms ESM-GearNet-INR-MC, it seems like structure pretraining is unnecessary if sequence+surface pretraining is conducted. While it shows the importance of the surface modality, the main point of the paper, doesn't it somewhat inconsistent with the overall nuance of the paper? As I recall it, the paper (especially within title, Table1, and introduction) emphasized the pre-training strategy incorporating all of sequences, structures, and surfaces.
> > - Although I agree the strengths of the work, I think it would be nice if the paper also discusses its weakness regarding the necessity of true protein structures. Having a known protein structures seems like a pretty strong restriction. I'm just thinking out loud here but, if we know the true structure of a protein, I assume there's a good chance that we might already know the answers for the downstream tasks or we have good non-ML approaches to obtain the answers.

---

> > > ### Author Response · Authors · 2023-11-22
> > > **Thank you. Our updates for post-rebuttal comments.**
> > >
> > > We thank the reviewer for the valuable feedback that helps us to improve our work. Please see our responses below:
> > >
> > > ---
> > >
> > > > C1: Considering that ESM-GearNet-INR outperforms ESM-GearNet-INR-MC, it seems like structure pretraining is unnecessary if sequence+surface pretraining is conducted. While it shows the importance of the surface modality, the main point of the paper, doesn't it somewhat inconsistent with the overall nuance of the paper? As I recall it, the paper (especially within title, Table1, and introduction) emphasized the pre-training strategy incorporating all of sequences, structures, and surfaces.
> > >
> > >
> > > **Response**
> > >
> > > We appreciate the insightful comment. While our additional results (ESM-GearNet-INR) confirm the importance of incorporating the surface modality in pre-training, our results suggests that identifying the better combination or strategy, especially between surface and structure pre-training, necessitates further exploration. We have added the necessity for investigating approaches to more effectively combine surface and structural pre-training in the manuscript.
> > >
> > > ---
> > >
> > > > C2: Although I agree the strengths of the work, I think it would be nice if the paper also discusses its weakness regarding the necessity of true protein structures. Having a known protein structures seems like a pretty strong restriction. I'm just thinking out loud here but, if we know the true structure of a protein, I assume there's a good chance that we might already know the answers for the downstream tasks or we have good non-ML approaches to obtain the answers.
> > >
> > > **Response**
> > >
> > > Thank you for the valuable comment. In concurrence with your viewpoint, we stated the limitation of our method in the absence of structure.

---

### Official Review · Reviewer_AyLE · 2023-10-30

**Soundness:** 3 good
**Presentation:** 2 fair
**Contribution:** 3 good
**Rating:** 6
**Confidence:** 2

**Summary:**

This paper introduced a protein multimodal representation learning by leveraging information from protein sequence, structure, and surfaces. Specifically, the authors proposed a model which uses INR to encode protein surface information by using point cloud representation and fused it into the structure embedding. Empirically the authors have shown the performance boost comparing to the original baseline methods which only use structure information.

**Strengths:**

1. This paper has shown that fusing protein surface data with protein structure data generates better protein representation that benefits various downstream tasks.
2. The proposed ProteinINR architecture is modularized and the encoders can be easily replaced by other implementations.
3. Various downstream tasks experiments are performed.
4. Clear description of training procedures and descriptions of the training data usage.

**Weaknesses:**

1. Lack of in-depth analysis on the effect of pre-training order. In Table 3, it is shown that GearNet-INR-MC performs better than GearNet-MC-INR for both $F_{max}$ and AUPR, however, no further analysis is provided.
2. INR seems only work well when both structure and sequence information are present. In Table 2, both GearNet-INR-MC and GearNet-INR have similar performance as their counterparts. Only ESM-GearNet-INR-MC shows relatively bigger improvements comparing to ESM-GearNet-MC.

**Questions:**

1. Why does pre-training order impact the EC tasks? Does this observation apply to other downstream tasks as well?
2. How does batchsize in finetuning affect the overall downstream performance?
3. In section 4.1.4, how is the clamp value decided?

---

> ### Author Response · Authors · 2023-11-20
> **Thank you. Our updates.**
>
> We thank the reviewer for the valuable feedback that helps us to improve our work. Please see our responses below:
>
> > W1: Lack of in-depth analysis on the effect of pre-training order. In Table 3, it is shown that GearNet-INR-MC performs better than GearNet-MC-INR for both  and AUPR, however, no further analysis is provided.
> >
> > W2: INR seems only work well when both structure and sequence information are present. In Table 2, both GearNet-INR-MC and GearNet-INR have similar performance as their counterparts. Only ESM-GearNet-INR-MC shows relatively bigger improvements comparing to ESM-GearNet-MC.
> >
> > Q1: Why does pre-training order impact the EC tasks? Does this observation apply to other downstream tasks as well?
>
> **Response**
>
> Table 3 correction: The experiment in Table 3 was designed to examine the influence of the order of structure and surface pre-training, thus we did not use sequence pre-training. We correct “Pre-training order” part by removing the “sequence” part. Also, the units (such as 0.896 and 0.889) have been revised to 89.6 and 88.9, respectively as in Table 2. The title of the Ablation study has been revised to  "Experiment on Pre-training Order." Additionally, we have conducted and added experimental results for other downstream tasks.
>
> Considering the original results (Table 2) and our additional experiment (ESM-GearNet-INR), it is evident that ESM has the most dominant impact on the downstream tasks. So, when the sequence pre-training information is present, structure and surface play a supportive role in enhancing performance. Table 3 enables us to compare the significance of structure and surface pre-training while excluding the dominant influence of sequence. Based on the results of GearNet-INR (466.1) and GearNet-MC (498.3) shown in Table 2, it can be concluded that in the absence of pre-training on sequences, structure pre-training has a greater influence on downstream tasks compared to surfaces. We conjecture that this observation supports the findings that the structure encoder (GearNet-INR-MC), which learned structure information last, had superior performance compared to GearNet-MC-INR, which was pre-trained in the opposite manner, as shown in Table 3.
>
> ---
>
> > Q2: How does batchsize in finetuning affect the overall downstream performance?
>
>
> **Response**
>
> We appreciate your constructive feedback. At the early stage of our research, we experimented with different batch sizes (e.g., 8, 16) for several tasks and found no notable differences in performance. So, we consistently conducted the experiments with a batch size of 16 for efficiency and fairness.
>
> ---
>
> > Q3: In section 4.1.4, how is the clamp value decided?
>
>
> **Response**
>
> We used the clamp value of 0.2, as employed in the DeepSDF paper (https://arxiv.org/abs/1901.05103). We added this sentence in the manuscript.
>
> ---

---

> > ### Comment · Reviewer_AyLE · 2023-11-22
> > **Post rebuttal comments and follow-up**
> >
> > I appreciate authors for their responses to my questions, and they have addressed my concerns. I have a follow-up question that is similar to what reviewer VAn9 brought up. It seems that the additional INR does not consistently improve the model's performance (when ESM is absent). Considering the response to my original review, it appears to me that INR and structure embedding are incongruent without the sequence model?

---

> > > ### Author Response · Authors · 2023-11-22
> > > **Thank you. Our updates for post-rebuttal comments.**
> > >
> > > We thank the reviewer for the valuable feedback that helps us to improve our work. Please see our responses below:
> > >
> > > > C1: I appreciate authors for their responses to my questions, and they have addressed my concerns. I have a follow-up question that is similar to what reviewer VAn9 brought up. It seems that the additional INR does not consistently improve the model's performance (when ESM is absent). Considering the response to my original review, it appears to me that INR and structure embedding are incongruent without the sequence model?
> > >
> > > **Response**
> > >
> > > We thank the thoughtful comment. Although the addition of surface pre-training without ESM doesn't result in a significant improvement , we have observed an increase in the average performance (GearNet-INR > GearNet, GearNet-INR-MC > GearNet-MC), confirming the importance of incorporating the surface modality in pre-training for protein representation learning. However, despite this, the additional experiment (ESM-GearNet-INR outperforming ESM-GearNet-INR-MC) and the relatively modest performance gains when comparing GearNet-MC and GearNet-INR-MC suggest that further investigation is needed to find the better combination or strategy for integrating all three modalities, particularly the effective integration of surface and structure pre-training. We have clearly stated the need to explore approaches to a better way of combining surface and structure pre-training in the manuscript.

---

### Official Review · Reviewer_1KWf · 2023-11-09

**Soundness:** 3 good
**Presentation:** 3 good
**Contribution:** 3 good
**Rating:** 6
**Confidence:** 4

**Summary:**

Proteins can be represented in many different ways (sequence, structure and surfaces). While the sequence and structure based representations have been explored a lot in the literature, surface based representations have not been explored. This paper proposes a pretraining strategy that incorporates all three modalities for protein representation learning. They propose Implicit Neural Representations to learn the surface characteristics of proteins.

**Strengths:**

- The proposed method is novel and is the first method to use surface based pretraining for proteins. This could open up further avenues of study.
- Incorporates several advances from computer vision (DeepSDF, DSPoint, KPConv, decoder by Lee et al., 2023 etc) into the protein domain.
- The evaluation is fair and is done against state of the art methods.
- Pretraining is done on a large number of structures
- The paper is well written and easy to follow.

**Weaknesses:**

- Performance of the method is more or less in line with existing work (ESM-Gearnet-MC), with any improvements being marginal for some tasks.
- Some more abalations/baselines would clarify the contribution of the surface based features  (see questions section).

**Questions:**

1. Why do you think is the performance so much dependent on the pretraining order (Table 3). Please clarify this in the paper.
2. Does seq->surface->3d structure outperform seq->structure->surface for all tasks? Table 3 only shows EC.
3. It would be interesting to see if we can avoid the structure encoder altogether by just relying on sequence and surface features.
4. In Table 1, it would be nice to have the performance on just ESM.

---

> ### Author Response · Authors · 2023-11-20
> **Thank you. Our updates.**
>
> We thank the reviewer for the valuable feedback that helps us to improve our work. Please see our responses below:
>
> > W2: Some more abalations/baselines would clarify the contribution of the surface based features (see questions section).
>
> **Response**
>
> Thank you for your comment. We additionally evaluated the performance of ESM-GearNet-INR, which did not undergo structure pre-training, on downstream tasks and observed the highest level of performance and added this result to Table 2. This serves as additional experimental evidence demonstrating the significance of surface features.
>
> ---
>
> > Q1: Why do you think is the performance so much dependent on the pretraining order (Table 3). Please clarify this in the paper.
>
> **Response**
>
> Table 3 correction: The experiment in Table 3 was designed to examine the influence of the order of structure and surface pre-training, thus we did not use sequence pre-training. We correct “Pre-training order” part by removing the “sequence” part. Also, the units (such as 0.896 and 0.889) have been revised to 89.6 and 88.9, respectively as in Table 2. The title of the Ablation study has been revised to  "Experiment on Pre-training Order." Additionally, we have conducted and added experimental results for other downstream tasks.
>
> Considering the original results (Table 2) and our additional experiment (ESM-GearNet-INR), it is evident that ESM has the most dominant impact on the downstream tasks. So, when the sequence pre-training information is present, structure and surface play a supportive role in enhancing performance. Table 3 enables us to compare the significance of structure and surface pre-training while excluding the dominant influence of sequence. Based on the results of GearNet-INR (466.1) and GearNet-MC (498.3) shown in Table 2, it can be concluded that in the absence of pre-training on sequences, structure pre-training has a greater influence on downstream tasks compared to surfaces. We conjecture that this observation supports the findings that the structure encoder (GearNet-INR-MC), which learned structure information last, had superior performance compared to GearNet-MC-INR, which was pre-trained in the opposite manner, as shown in Table 3.
>
> ---
>
> > Q2: Does seq->surface->3d structure outperform seq->structure->surface for all tasks? Table 3 only shows EC.
>
> **Response**
>
> Thanks for the helpful comment. We conducted additional experiments on other tasks and found the similar trends with EC task. The results were added in Table 3.
>
> ---
>
> > Q3: It would be interesting to see if we can avoid the structure encoder altogether by just relying on sequence and surface features.
>
> **Response**
>
> We appreciate your insightful suggestion. Currently, our codes are integrated in such a way that the structural encoder is included in the INR-training code, which poses difficulties in carrying out the recommended experiments throughout the rebuttal time. We value your insightful comments and will contemplate investigating these experiments in our forthcoming research. Meanwhile, we have performed examinations on pre-training using INR after ESM only with exclusion of structure pre-training.
>
> ---
>
> > Q4: In Table 1, it would be nice to have the performance on just ESM.
>
> **Response**
>
> Thank you for your comment. We added ESM-2 (pre-trained on only sequences) results, which are extracted from the reference paper (https://arxiv.org/abs/2301.12040), as a baseline in Table 2.
>
> ---

---

### Meta-Review · Area_Chair_hwmp · 2023-12-06

**Metareview:**

The paper considers protein representation learning and proposes approach that integrates protein sequence, structure and surface information. The proposed approach is novel and the presentation is clear.  The AC and reviewers appreciate the author feedback which clarifies many points. It would be great if the authors could incorporate their final remarks to reviewer  rs2T as they shed valuable light on the performance of the proposed approach vs. ESM2.

**Justification For Why Not Higher Score:**

The novelty is not that great and the advantage of the proposed approach is not that impressive, with cases where baselines perform better or the advantage is minimal.

**Justification For Why Not Lower Score:**

The approach is the first to combine sequence structure and surface information in protein representation learning and will certainly inspire valuable future work.

---

### Decision · Program_Chairs · 2024-01-16

Accept (poster)